# Analyzing the Performance of Millimeter Wave MIMO Antenna under Different Orientation of Unit Element

**DOI:** 10.3390/mi14111975

**Published:** 2023-10-24

**Authors:** Tanvir Islam, Fahad N. Alsunaydih, Fahd Alsaleem, Khaled Alhassoon

**Affiliations:** 1Department of Electrical and Computer Engineering, University of Houston, Houston, TX 77204, USA; tislam7@cougarnet.uh.edu; 2Department of Electrical Engineering, College of Engineering, Qassim University, Unaizah 56452, Saudi Arabia; f.alsunaydih@qu.edu.sa (F.N.A.); f.alsaleem@qu.edu.sa (F.A.)

**Keywords:** MIMO antenna, unit element, isolation, ECC, 28 GHz, 5G

## Abstract

In this paper, a compact and simplified geometry monopole antenna with high gain and wideband is introduced. The presented antenna incorporates a microstrip feedline and a circular patch with two circular rings of stubs, which are inserted into the reference circular patch antenna to enhance the bandwidth and return loss. Roger RT/Duroid 6002 is used as the material for the antenna, and has overall dimensions of W_S_ × L_S_ = 12 mm × 9 mm. Three designs of two-port MIMO configurations are derived from the reference unit element antenna. In the first design, the antenna element is placed parallel to the reference antenna, while in the second design, the element is placed orthogonal to the reference element of the antenna. In the third design, the antenna elements are adjusted to be opposite each other. In this study, we analyze the isolation between the MIMO elements with different arrangements of the elements. The MIMO configurations have dimensions of 15 mm × 26 mm for two of the cases and 15 mm × 28.75 mm for the third case. All three MIMO antennas are made using similar materials and have the same specifications as the single element antenna. Other significant MIMO parameters, including the envelope correlation coefficient (ECC), diversity gain (DG), channel capacity loss (CCL), and mean effective gain (MEG), are also researched. Additionally, the paper includes a table summarizing the assessment of this work in comparison to relevant literature. The results of this study indicate that the proposed antenna is well-suited for future millimeter wave applications operating at 28 GHz.

## 1. Introduction

The recent advancements in communication systems (shifting from 4G to 5G, and currently from 5G to 6G) have brought about numerous changes and revisions. These changes have been made to meet the requirements, which are a basic necessity for operating over future wireless communication spectrum [1,2]. These requirements are aimed at achieving a high data rate, increasing the throughput and data capacity, as well at creating low-cost compact devices. Due to changes in designing communication models, the requirements for changing the communication entities have also changed [3,4,5]. One of the most important parts of communication models is the antenna, and due to the aforementioned requirements and changes, the requirements for antennas have also changed [6,7].

High gain and wideband antenna have gained increasing attention in recent years due to the evolving demands in communication systems [8,9]. Devices that offer high data rates and cater to a sea of users, while being small and portable, are in in high demand. Antennas with features such as high gain, wideband, and compact, low-profile designs, have become crucial components in communication devices [10,11,12,13,14]. 

The implementation of multiple-input multiple-out (MIMO) antenna technology increases the capacity and reduces interference [15,16]. Some examples of usage of MIMO antenna system in future communication is given in Figure 1. By applying MIMO antennas, one can fulfil the requirements of present-day communication systems, as various techniques are used to improve the parameters [17]. In the literature, various antennas designs have been presented, many of which incorporate elements such as parasitic patches, Electronic Band Gaps (EBGs), metamaterials, Frequency Selective Surfaces (FSS) layers, and Dielectric Resonator Antenna (DRA) to improve antenna isolation [18,19,20]. Numerous MIMO antennas are self-decoupled, do not contain any structure or layer, and operate under a desired range of isolation [21,22].

It is mentioned in [23] that a two-port MIMO antenna with a modest overall size of 12 mm × 24 mm × 0.79 mm was developed. The antenna has a wide band coverage of 28.2 to 30.7 and an Envelop Correlation coefficient (ECC) of 0.001. The investigation of significant MIMO properties such isolation, Diversity Gain (DG), Mean Effective Gain (MEG), and gain constitutes the exclusive focus of this paper. Ref. [24] investigated an additional tiny 15 mm × 25 mm two-port MIMO antenna. The antenna covers the 26.5 to 29.5 wideband frequency range and has a peak gain of 5.8 dBi. An array MIMO antenna with four ports and a size of 17.2 mm × 62 mm is presented in Ref. [25]. The antenna offers high gain due to array technology (around 13.6 dBi) and offers a wideband capability of 26.6–30.2 GHz. The antenna is large in size due to placing antenna element in parallel.

Etching slots from the ground plane can also increase an antenna’s bandwidth and gain if it has a defective ground structure (DGS) [26]. In [27], a wideband antenna with a gain of 5.42 GHz and an operational band width of 26–32 GHz is examined. The antenna provides an isolation of 35 dB and an ECC of about 0.005. The DGS ground plane enhances the bandwidth characteristics of the antenna. Another DGS ground plane antenna with a bandwidth of 26.2–30 GHz and overall dimensions of 30 mm × 35 mm × 0.76 mm is presented in [28]. The antenna’s strong gain of 8.3 dBi is a result of both the array’s structure and DGS. 

A compact sized 24 mm × 20 mm, wideband 33–44 GHz, and transparent antenna is reported in [29]. The antenna recorded a relative permittivity of 2.3 on the transparent substrate Plexiglass. The antenna is small and broad, but its ECC and isolation values are poor (0.1 and 1.6 dB, respectively). Ref. [30] present an antenna that provides good isolation at a cost of about 45 dB. The antenna is 47.4 mm × 32.5 mm and has a broad frequency range of 36.8 to 40 GHz. The antenna structure is zigzag with large dimensions, which cannot be fixed in any communication model for practical applications. The MIMO antenna for mobile communication is presented in [31]. The antenna’s enormous size of 158 mm by 77.8 mm and bandwidth of 25 to 40 GHz are also advantageous. The antenna’s gain is 7.2 dBi; however, its ECC and isolation values of 0.5 and 17 dB, respectively, are not sufficient.

For 28 GHz applications, a two-port MIMO antenna is suggested in [32]. The antenna is 20.5 mm by 12 mm overall, operates between 26.5 and 30 GHz, and has a peak gain of 8.75 dBi. Table 1 contain the comparison of proposed antenna along with state of the art. The performance of the antenna is studied after placing the MIMO element of the antenna in various orientations and this found that the MIMO antenna offers good results when two elements are orthogonal to each other. The limitation of this work is that only the transmission and reflection coefficient is studied. This study and discussion focus on the performance of two-port MIMO antennas in terms of transmission and reflection coefficients (ECC, Channel Capacity Loss (CLL), DG, and MEG).

Three sections make up the remaining portion of this study. The unit element of an antenna is examined in the second section, along with its findings. The three scenarios of the recommended MIMO antenna are examined in Section 3, along with a hardware prototype and comparison chart. This work is concluded with references in Section 4. Furthermore, the following improvements and developments are obtained from the proposed design:Compact design with low structural complexity;Analysis of the MIMO antenna under various element orientation;Low mutual coupling and ECC from all three MIMO designs;High gain antenna without using additional layers.

## 2. Design and Results of the Unit Element of the Antenna

The suggested antenna’s layout is illustrated in Figure 2. In this design, a substrate made of Rogers RT/Duroid 6002 with a loss tangent of 0.0012 and a relative permittivity of 2.94 is employed. The antenna measures 12 mm × 9 mm × 1.52 mm overall (LS × WS × H). It is clear that the antenna geometry has a straightforward design with a monopole antenna, a microstrip feedline, and a circular patch with several circular rings. The antenna is filled with numerous circular rings to maximize bandwidth and reduce return loss. The antenna’s reverse side has a full ground plane with a 0.035 mm copper sheet. The High Frequency Structural Simulator (HFSS v 19) software application is used to study the results of the presented work. Below are the optimized parameters:

### 2.1. Designing Steps

The three steps of the design evaluation for the antenna are illustrated in Figure 3. Using the equation provided in [33,34,35], a circular patch antenna for 28 GHz is built in the first stage. At 27.8 GHz, the antenna is in use. Then, a circular ring with an outside radius of 3 mm and an inner radius of 2.5 mm is put into the antenna. The antenna’s performance improved in terms of bandwidth and return. The figures show that there has been an improvement in return loss from −12 dB to −22 dB. In the third stage of the design, another circular ring is inserted into the antenna, which has an inner and outer radius of R_3_ and R_4_. This procedure significantly reduced the antenna’s return loss, enabling wideband operation. The resulting antenna, shown in Figure 3, has a wideband of 8.5 GHz and a return loss of −28 dB.

### 2.2. Outcomes of the Unit Element

The comparison between the predicted and tested S_11_ parameter of the recommended antenna single element is given in Figure 4. It is shown in the figure that the antenna operates over a ultrawide band with an impedance bandwidth of 26–34.25 GHz (S_11_ < −10 dB). The proposed design offers two resonant frequencies around 28 GHz and 31.75 GHz with a return loss of 28 dB. From the figure it can also be seen that the S_11_ plot generated from the software and tested is quite similar with minor distortions. Additionally, Figure 4 also includes the hardware prototype created to test the results of the simulation. 

The suggested antenna’s radiation pattern at specific frequencies of 28 GHz and 32 GHz is shown in Figure 5. For both resonating frequencies, the recommended antenna provides a broad side radiation pattern in the theoretical H-plane and a slightly distorted radiation pattern in the E-plane. The manipulation in E–plane is perhaps due to multiple circular ring stub loading. Moreover, the measured radiation pattern shows similarity with the simulated pattern, which can be seen in the figure below.

The gain vs. frequency plot of this design is provided in Figure 6. It can be seen from the figure that the antenna gives a high gain of more than 11 dBi through the operational region of 26–32.75 GHz. The peak value of gain is noticed at 27.85 GHz and 31.75 GHz with the value of 11.5 dBi and 11.75 dBi, respectively. Moreover, the tested value of gain is also added to this figure to provide a comparison with simulated results. Additionally, Figure 5 also provides information on the antenna’s radiation efficiency. It can be shown that antennas use the operating spectrum to provide a radiation efficiency of over 93%. A high value of radiation efficiency of around 94% and 96% is observed at resonance frequencies of 27.85 GHz and 31.75 GHz.

## 3. Two-Port MIMO Antenna

This section studies and discusses the idea for the antenna’s two-port MIMO architecture. In order to analyze the MIMO characteristics of closely spaced elements, three separate scenarios are used. In the first scenario, both parts are positioned side by side. One MIMO antenna component is orthogonal to the other in the second scenario. The element is parallel but positioned on the opposite side in the third instance. The hardware prototypes are fabricated and tested for all aforementioned cases. 

To measure the reflection and transmission coefficient of the antenna, a vector network analyzer (VNA), 220 ZVA by Rohde & Schwarz, is used. To measure the far field, the antenna is placed in a newly designed shielded millimeter-wave anechoic chamber and we utilize a multi probe array technique, which provides accurate result for a spanning angle of 180° [36].

### 3.1. Case 1: Parallel Placed MIMO Elements

In Figure 7, a two-port MIMO antenna’s structural geometry and hardware prototype is shown. As can be observed, there is a space of S_1_ = 4.75 mm between the second element and the reference element. The entire size of the MIMO antenna in this instance is M_XI_ × M_Y1_ = 15 mm × 26 mm. The same substrates and other design criteria were employed for one element only. To validate the simulated results of the antenna, the hardware prototype is created.

In Figure 8, the measured and predicted S-parameter of a two-port MIMO antenna with both elements parallel is given. The figure shows that the antenna provides a bandwidth of 26.2–34.5 GHz (S11 < −10 dB). In this instance, a two-port MIMO antenna with a return loss of less than 28 dB was developed to be resonant on 28 GHz and 31.5 GHz. Furthermore, the transmission coefficient is also examined and provided in Figure 8 in order to evaluate mutual interaction between MIMO antenna elements. The provided antenna performs at less than the desired level of isolation (−20 dB). It is clear that the antenna provides isolation of about 25 dB across the operational bandwidth, with peak values of 35 dB at 28 GHz and 30 GHz. Furthermore, the proposed antenna is suited for emerging millimeter wave applications due to the remarkable agreement between the measured and simulated findings.

Our study uses the MIMO parameters, including the ECC, DG, CCL, and MEG. To examine the performance of single unit element in MIMO configuration, ECC is studied. DG is the study of losses in the form of transmission, and CCL is the examination of correlation losses in MIMO systems. The MEG is analyzed to study the power received in a fading area. The mathematical equations below, which are used to calculate these MIMO parameters, are well explained in [37,38,39,40].
(1)$$ECC=\rho_{eij}=\frac{{\left|S_{ii}*S_{ij}+S_{ji}*S_{jj}\right|}^{2}}{\left(1-{\left|S_{ii}\right|}^{2}-{S_{ij}}^{2})(1-{\left|S_{ji}\right|}^{2}-{S_{jj}}^{2}\right)}$$
(2)$$DG=10\sqrt{{1-|\rho ij|}^{2}}$$
(3)$$MEGi=0.5\left(1-\sum_{i=1}^{N}\left|Sij\right|\right)$$
(4)$$CCL=-\log_{2}det\left|\psi^{R}\right|$$
where *ψ^R^* refers to the below matrix for receiving antenna correlation.
(5)$$\psi^{R}=\left[\begin{array}{cccc} \rho 11 & \rho 12 & \rho 13 & \rho 14\\ \rho 21 & \rho 22 & \rho 23 & \rho 24\\ \rho 31 & \rho 32 & \rho 33 & \rho 34\\ \rho 41 & \rho 42 & \rho 43 & \rho 44 \end{array}\right]$$

The ECC and DG of the suggested parallel element placed design are shown in Figure 9a. It can be seen from the figure that the antenna offers ECC < 0.0015 throughout the operational region. In an ideal case, the ECC should be equal to zero, but in the proposed design the value of ECC is approaching zero. In the same figure, the DG is also given. The antenna offers DG > 9.8 dB over all operational bandwidth of 26.2–34.5 GHz. The MIMO antenna studied in this paper offers CCL < 0.025 bits/Hz/s, which is under the acceptable range. The MEG is also provided in Figure 9b, and it is observed that the antenna offers MEG around −6.3 dB. Moreover, the tested results are also added, which shows strong agreement with the simulated results. The results prove that the proposed dual port antenna can be considered as a good applicant for future millimeter wave application. 

### 3.2. Case 2: Orthogonally Placed MIMO Elements

Figure 10 shows the structural geometry and hardware prototype for a two-port MIMO antenna. As can be seen, the second antenna element is positioned 90 degrees apart from the reference element in an orthogonal position. Two elements must be separated by a predetermined distance of S_2_ = 4 mm. The entire size of the MIMO antenna in this instance is M_X2_ × M_Y2_ = 15 mm × 28.75 mm. The same substrates and other design criteria were employed for one element only. To validate the simulated results of the antenna, the hardware prototype is created.

In Figure 11, the measured and predicted S-parameter of two-port MIMO antenna with both elements placed orthogonally is given. Figure illustrates that a two-port MIMO antenna can transmit data at a bandwidth of 26–34.75 GHz (S11 −10 dB). In this particular case, a two-port MIMO antenna with a return loss of less than 30 dB was developed to be resonant on 28 GHz and 31.5 GHz. Furthermore, the transmission coefficient is also examined and provided in Figure 10 in order to evaluate mutual interaction between the MIMO antenna elements. The provided antenna performs at less than the desired level of isolation (20 dB). With peak values of 42 dB at 28 GHz and 30 GHz, the antenna delivers isolation of 32 dB over the operational bandwidth. Furthermore, the proposed antenna is suited for upcoming millimeter wave applications due to the remarkable agreement between the measured and simulated findings.

Similarly to the above antenna presented in Section 3.1, we also study the important MIMO parameters for this antenna. In Figure 12a, ECC and DG are given, and in Figure 12b, CCL and DG are given. It is notable that the proposed orthogonal placed antenna element design offers ECC < 0.0001 throughout the operational region. The antenna offers DG around 9.99 dB over an operating bandwidth of 26–34.75 GHz. The MIMO antenna studied in this section offers CCL < 0.001 bits/Hz/sec, which is under the acceptable range. A further significant variable is the mean effective gain (MEG), which is shown in Figure 12b. It has been found that the antenna provides MEG at 6.25 dB. The tested results are also included, which exhibit excellent agreement with the predicted results. The outcome shows that the suggested two-port antenna qualifies as a strong candidate for upcoming millimeter wave applications. 

### 3.3. Case 3: Parallel Placed Opposite to Each Other MIMO Elements

Figure 13 shows the two-port MIMO antenna’s organizational layout and hardware prototype. As can be seen, the second antenna element in this instance is positioned parallel to the reference element—but on the opposite side. The distance between the MIMO antenna’s two elements is S_3_ = 5.2 mm. The MIMO antenna in this example has the same overall size as the antenna in the previous scenario, which is M_X3_ × M_Y3_ = 15 mm × 26 mm. The location of the antenna with the reference antenna is the only distinction between cases 3 and 1. To validate the simulated results of the antenna, the hardware prototype is created. 

In Figure 14, the measured and predicted S-parameter of two-port MIMO antenna with both elements parallel is given. The figure shows that the antenna provides a bandwidth of 26.5–34.5 GHz (S11 −10 dB). The two-port MIMO antenna in this instance has a return loss of less than 28 dB with resonances at 27.9 GHz and 31.75 GHz. Furthermore, the transmission coefficient is also examined and provided in Figure 14 in order to evaluate mutual interaction between MIMO antenna elements. The provided antenna performs at less than the desired level of isolation (20 dB). With peak values of 38 dB at 27.9 GHz and 31.75 GHz, the antenna delivers isolation of 28 dB over the operational bandwidth. Furthermore, the proposed antenna is suitable for upcoming millimeter wave applications due to the remarkable agreement between the measured and simulated findings.

As in the previous two designs, the important MIMO parameters of this antenna are also analyzed. In Figure 15a,b the ECC, DG, CCL, and MEG of the proposed parallel element placed design is given. The figure shows that the antenna provides ECC < 0.0015 in the operating frequency band. Over an operational bandwidth of 26.2–34.5 GHz, the antenna provides DG > 9.78 dB. The most crucial MIMO parameter is the channel capacity loss. The CCL of the MIMO antenna investigated in this research, which is under an acceptable range, is 0.035 bits/Hz/sec. Figure 15b also shows the mean effective gain (MEG), which shows that the antenna provides MEG at a level of about −6.38 dB. Moreover, the tested results are also added, which shows strong agreement with the simulated results. The results prove that the proposed dual port antenna can be considered as a good applicant for future millimeter wave applications.

Figure 16 presents the surface current distribution of the various cases of the MIMO antenna. It can be seen that in all configurations there is a low amount of current induced in element-2 when element-1 is excited, which results in low coupling among MIMO elements. Moreover, it can be seen from Figure 16b that the minimum amount of current is induced in case 2, thus the surface current analysis also verifies the results presented in previous sections. Table 2 summarizes the comparison among various parameters of all three configurations.

## 4. Conclusions

For 5G applications, a wideband, small, simple, high gain, and high radiation efficiency antenna is suggested in this research. For 28 GHz applications, a circular patch antenna was first developed. Later, to acquire the wideband, circular rings are placed into the antenna. The antenna provides a high gain of 11.25 dBi and a wide band of 8.25 GHz. Afterward, three MIMO antenna were adopted from initial reference antenna. The aim when designing a MIMO antenna under various arrangements of elements is to analyze and study the performance of the antenna. It is studied and concluded from the results that the antenna offers wideband and acceptable value of isolation in all three cases. The results in terms of MIMO parameters are also acceptable values, as the antenna offers ECC around 0.001, DG 9.9 dB, CCL 0.001 bits/Hz/sec, and MEG around 6.38 dB. The performance of MIMO antenna defined in three cases is concluded in Table 2, where it can be seen that case 2 provides better minimum isolation of −30 dB, with ECC of 0.001 and DG of 9.99 dB. All the three cases offer values within the acceptable range, but case 2 offers better value compared to case 1 and case 3. Furthermore, the results and comparisons with existing literature demonstrate that our proposed work is the most suitable for future high gain and compact devices operating at millimeter wave frequencies.

## Figures and Tables

**Figure 1 micromachines-14-01975-f001:**
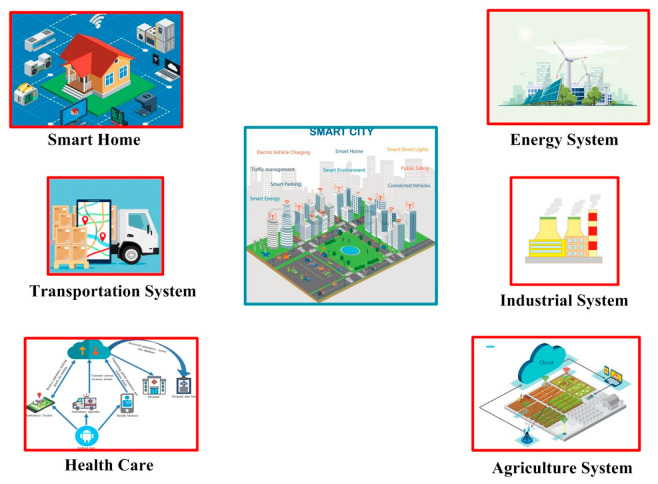
Practical applications of the MIMO antenna in the different domains of future communication systems.

**Figure 2 micromachines-14-01975-f002:**
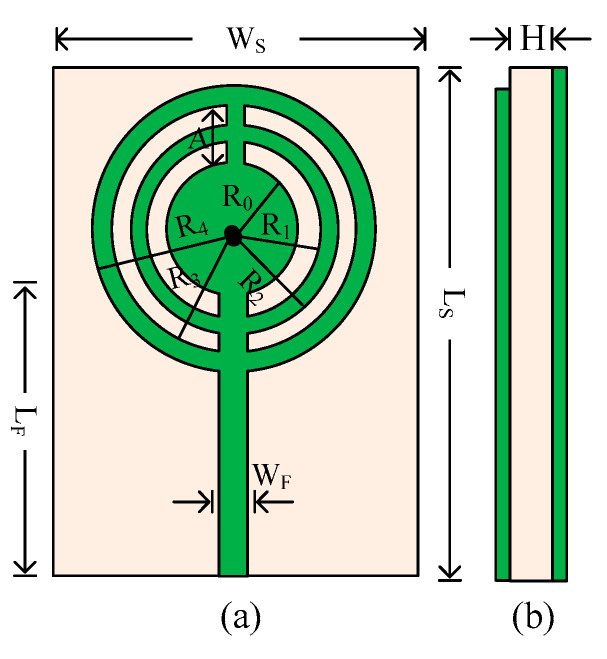
Structural layout of the recommended circular patch antenna. (**a**) Front view; (**b**) side view. LS = 12; WS = 9; L_F_ = 7.5; R_0_ =2; R_1_ = 2.5; R_2_ = 3; R_3_ = 3.5; R_4_ = 4.5; W_F_ = 0.75; A = 1.5; H = 0.79 (units in mm).

**Figure 3 micromachines-14-01975-f003:**
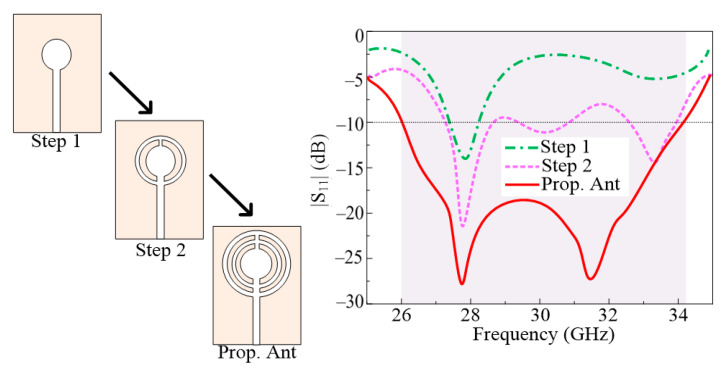
Design stages of the suggested antenna along with a |S_11_| plot.

**Figure 4 micromachines-14-01975-f004:**
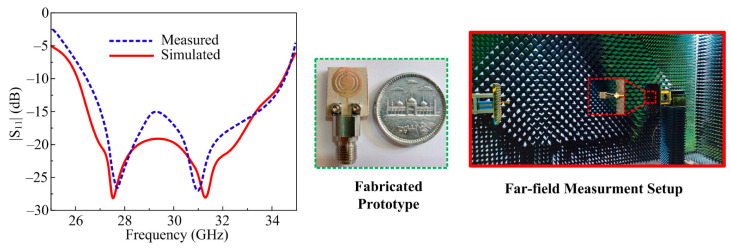
Predicted and tested S_11_ results of the proposed design along with antenna prototype and measurement setup.

**Figure 5 micromachines-14-01975-f005:**
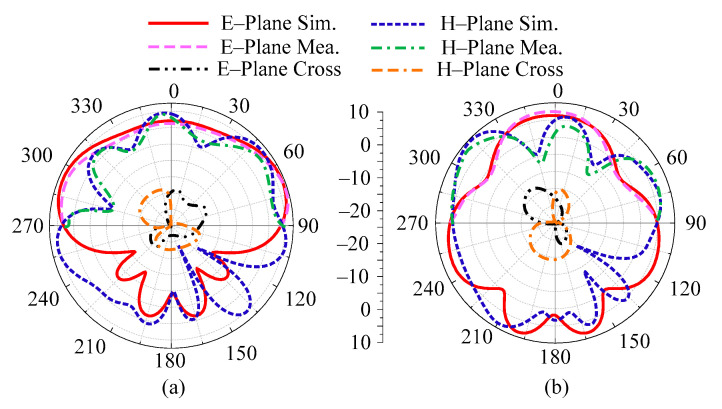
Predicted and tested radiation pattern of the proposed design at (**a**) 28 GHz and (**b**) 32 GHz.

**Figure 6 micromachines-14-01975-f006:**
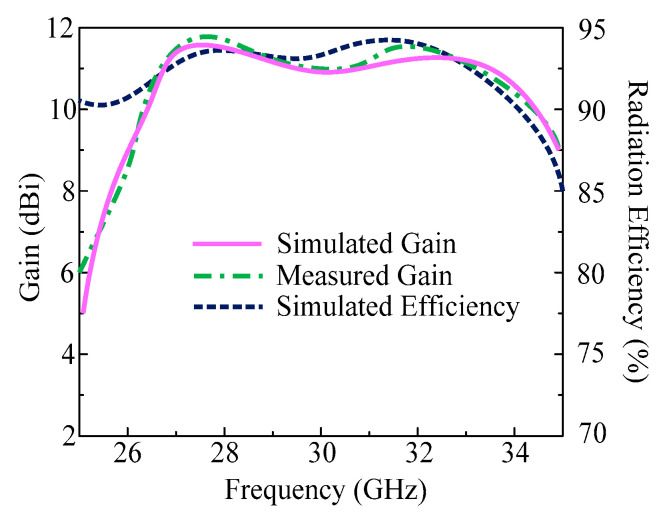
Predicted and tested gain of the proposed design along with radiation efficiency.

**Figure 7 micromachines-14-01975-f007:**
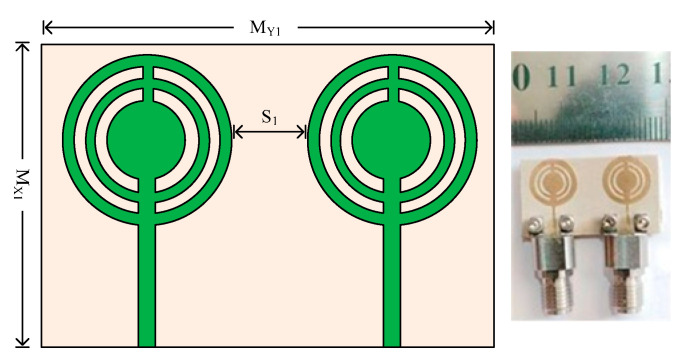
Geometry and structure of two-port MIMO antenna case 1 along with the prototype.

**Figure 8 micromachines-14-01975-f008:**
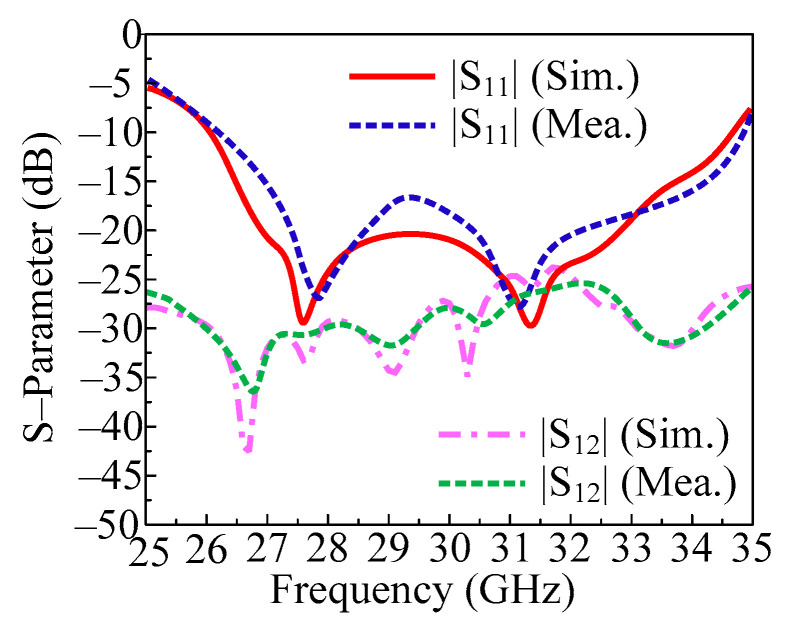
Predicted and tested S-parameter of the proposed MIMO antenna by placing the element parallel.

**Figure 9 micromachines-14-01975-f009:**
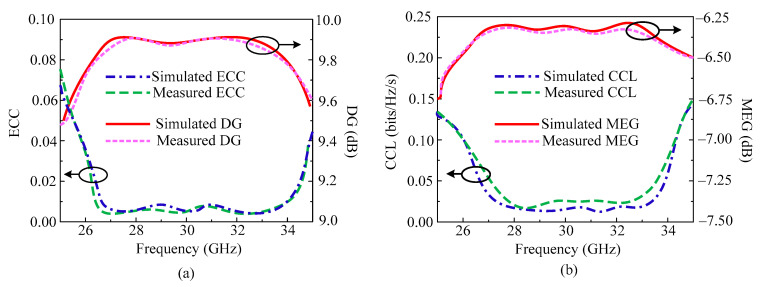
Predicted and tested MIMO parameters of parallel placed element MIMO design. (**a**) ECC and DG; (**b**) CCL and MEG.

**Figure 10 micromachines-14-01975-f010:**
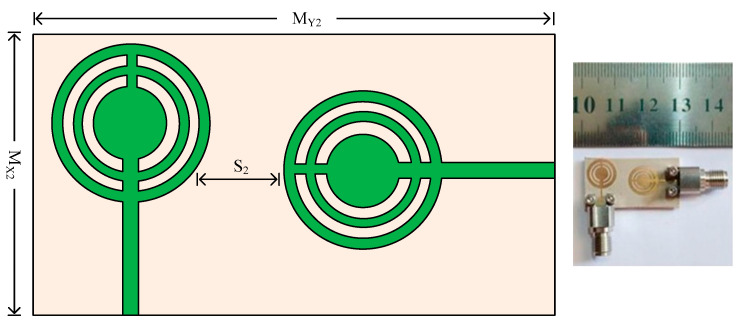
The geometry and structure of the two-port MIMO antenna for case 2 along with the prototype.

**Figure 11 micromachines-14-01975-f011:**
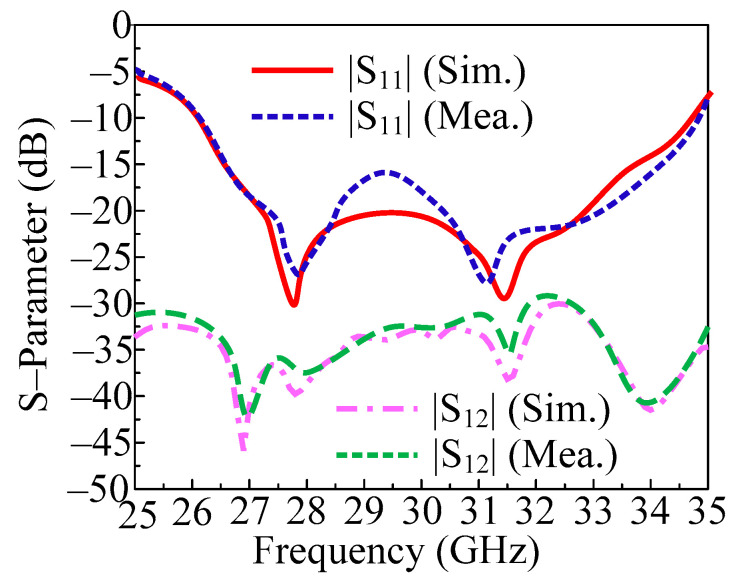
Predicted and tested S-parameter of the proposed MIMO antenna by placing the element orthogonal.

**Figure 12 micromachines-14-01975-f012:**
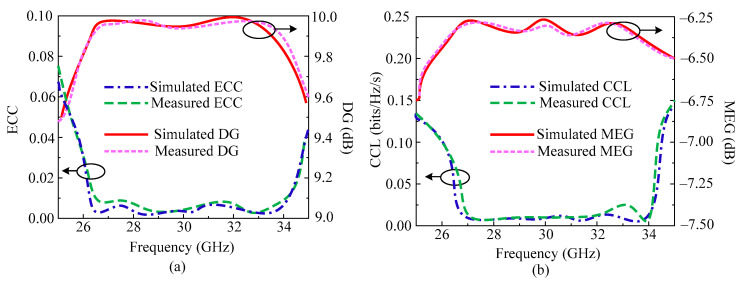
Predicted and tested MIMO parameters of orthogonal placed element MIMO design (**a**) ECC and DG (**b**) CCL and MEG.

**Figure 13 micromachines-14-01975-f013:**
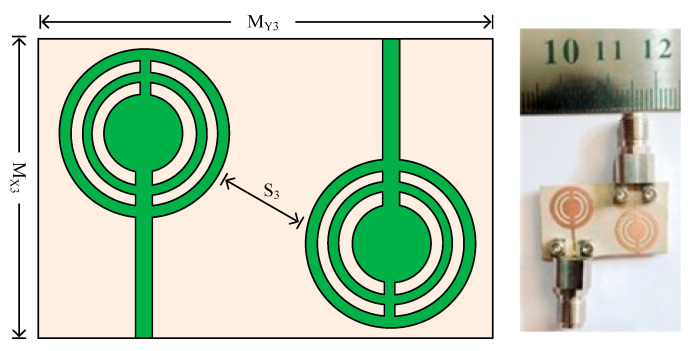
Geometry and structure of two-port MIMO antenna case 3 along with prototype.

**Figure 14 micromachines-14-01975-f014:**
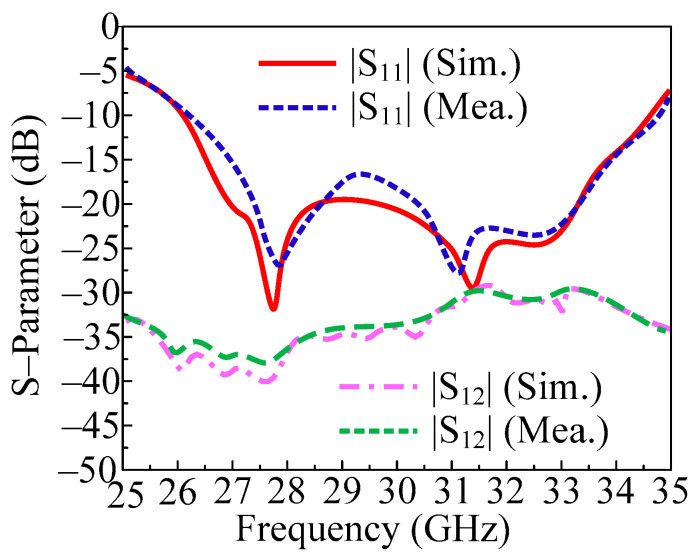
Predicted and tested S-parameter of the proposed MIMO antenna by placing the element parallel but on opposite side.

**Figure 15 micromachines-14-01975-f015:**
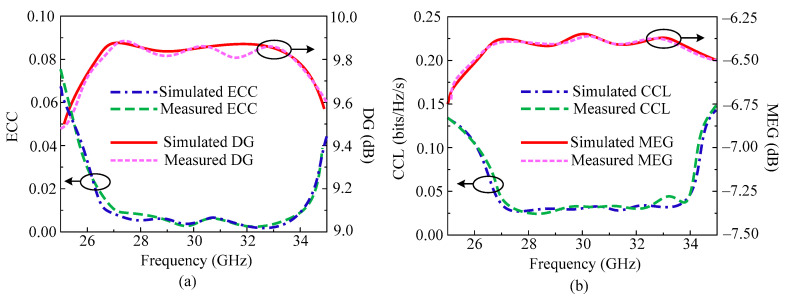
Predicted and tested MIMO parameters of parallel placed from opposite side element MIMO design. (**a**) ECC and DG; (**b**) CCL and MEG.

**Figure 16 micromachines-14-01975-f016:**
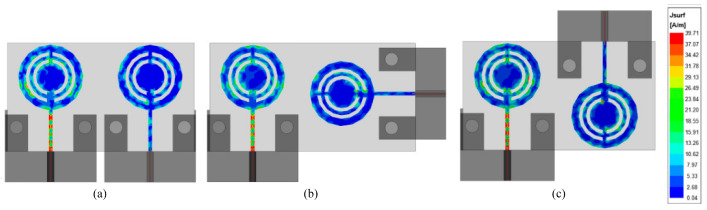
Surface current distribution of the various configuration at 28 GHz. (**a**) Case 1; (**b**) case 2; (**c**) case 3.

**Table 1 micromachines-14-01975-t001:** Comparison between the proposed work and existing work in the literature.

Ref.	Antenna Size (mm × mm × mm)	Bandwidth (GHz)	Peak Gain (dBi)	ECC	Isolation (dB)	Efficiency (%)
[23]	12 × 24 × 1.79	28–30.7	-	0.001	-	Not given
[24]	15 × 25 × 0.203	26.5–29.5	5.8	0.005	30	>87
[25]	17.2 × 62 × 0.8	26.6–30.2	13.6	0.001	35	Not given
[26]	30 × 15 × 0.25	26–30	5.42	0.005	35	>85
[28]	30 × 35 × 0.76	26.2–30	8.3	0.01	45	>82
[29]	24 × 20 × 1.85	33–44	-	0.1	16	>81
[30]	47.4 × 32.5 × 0.51	36.8–40	6.5	0.001	45	>79
[31]	158 × 77.8 × 0.381	25–40	7.2	0.5	17	>86
[32]	20.5 × 12 × 0.79	26.5–30	8.75	-	38	>89
This Work	15 × 26 × 1.52 15 × 26 × 1.52 15 × 28.75 × 1.52	26–34.25	11.25	<0.001	<38	>91

**Table 2 micromachines-14-01975-t002:** Comparison between outcomes of the recommended MIMO antenna discussed in various cases.

Antenna Type	Minimum Isolation (dB)	Envelope Correlation Coefficient	Channel Capacity Loss Bits/Hz/s	Mean Effective Gain (dB)	Diversity Gain (dB)
Case 1	−25	0.0015	0.025	−6.3	9.90
Case 2	−30	0.0010	0.001	−6.25	9.99
Case 3	−28	0.0015	0.035	−6.38	9.8

## Data Availability

All the data is available in the study.

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
