# Peer review of "Analyzing the Performance of Millimeter Wave MIMO Antenna under Different Orientation of Unit Element"

_micromachines, 2023, doi:10.3390/mi14111975_

Round 1
Reviewer 1 Report
The paper presents a compact and simplified monopole antenna design that incorporates circular stubs for bandwidth enhancement and performance improvement. The following comments aim to enhance the clarity and effectiveness of the manuscript:
1. The three distinct layout configurations explored in the study offer a significant insight into the antenna's performance under various arrangements. It is recommended to elaborate on the advantages and disadvantages of each configuration in terms of isolation, gain, and other relevant parameters. In the "Conclusion" section, the authors could provide a comparative analysis of these configurations and suggest the preferred layout, supported by insights from Table 2. Additionally, consider incorporating the comparative analysis within the "Two Port MIMO Antenna" section for better coherence.
2. The S11 plot in Figure 3 is crucial for demonstrating the antenna's impedance bandwidth. To enhance clarity, consider adding annotations to highlight the -10 dB impedance bandwidth starting point at 26 GHz.
3. Similar to the gain, Figures 9 and 15 should correspond accurately with the statement regarding the directive gain (DG) values over the specified operational bandwidth.
4. Clarify the rationale behind the discrepancy in value between A (4 mm), R3 (3.5 mm), and R0 (2 mm) in Figure 2. A more detailed explanation or correction might be necessary to ensure consistency and accuracy.
5. The observed similarity between simulated and measured radiation patterns in Figure 5 raises concerns. Provide additional details on the measurement setup and process to validate the accuracy of the measurements. If there are discrepancies, address them and discuss potential sources of error.
6. Gain and Efficiency Verification (Fig. 6): The high agreement between measured and simulated gain and radiation efficiency in Figure 6 requires further verification. Elaborate on the measurement methodology and any potential calibration procedures that were used to achieve this level of consistency.
7. As suggested, adding current distribution figures for the three layout configurations could provide valuable insights into the differences between them. This visual representation would enrich the understanding of the antenna's behavior.
8. The novelty of the proposed antenna is limited.
Moderate editing of English language is required.
Author Response
Reviewer no. 1
Comment no. 1: The paper presents a compact and simplified monopole antenna design that incorporates circular stubs for bandwidth enhancement and performance improvement. The following comments aim to enhance the clarity and effectiveness of the manuscript:
The three distinct layout configurations explored in the study offer a significant insight into the antenna's performance under various arrangements. It is recommended to elaborate on the advantages and disadvantages of each configuration in terms of isolation, gain, and other relevant parameters. In the "Conclusion" section, the authors could provide a comparative analysis of these configurations and suggest the preferred layout, supported by insights from Table 2. Additionally, consider incorporating the comparative analysis within the "Two Port MIMO Antenna" section for better coherence.
Author response: Authors appreciate the respected reviewer for their efforts and reviewing the manuscript. The authors have revised the manuscript, table 2 is provided having comparative analysis among all three configurations, based upon the results the case-2 is offering more promising results. The conclusion section is also updated by adding more details about the best configuration, moreover, the changes are highlighted in the revised manuscript.
Comment no. 2: The S11 plot in Figure 3 is crucial for demonstrating the antenna's impedance bandwidth. To enhance clarity, consider adding annotations to highlight the -10 dB impedance bandwidth starting point at 26 GHz.
Author response: Following the recommendation of the respected reviewer Fig 3 had been updated in the revised manuscript.
Comment no. 3: Similar to the gain, Figures 9 and 15 should correspond accurately with the statement regarding the directive gain (DG) values over the specified operational bandwidth.
Author response: The manuscript had been updated following the constructive comments of the respected reviewer.
Comment no. 4: Clarify the rationale behind the discrepancy in value between A (4 mm), R3 (3.5 mm), and R0 (2 mm) in Figure 2. A more detailed explanation or correction might be necessary to ensure consistency and accuracy.
Author response: Authors have cross verified the dimensions of the various parameters of the antenna, and it is found that there was a typo mistakes in the parameters value which are now updated in the revised manuscript.
Comment no. 5: The observed similarity between simulated and measured radiation patterns in Figure 5 raises concerns. Provide additional details on the measurement setup and process to validate the accuracy of the measurements. If there are discrepancies, address them and discuss potential sources of error.
Author response: Authors appreciate the respected reviewer for constructive comments. Detailed information about measurement setup is added and highlighted in section 3 of the revised manuscript. The high accuracy is the result of a newly developed measurement setup that utilizes multiprobe array technique to achieve results in full scanning angle of 3600. A research article explaining the methodology of measurement setup and its working is cited so that readers can refer to the article to have in-depth details about the setup.
Comment no. 6: Gain and Efficiency Verification (Fig. 6): The high agreement between measured and simulated gain and radiation efficiency in Figure 6 requires further verification. Elaborate on the measurement methodology and any potential calibration procedures that were used to achieve this level of consistency.
Author response: Detailed information about measurement setup is added and highlighted in section 3 of the revised manuscript. The high accuracy is the result of a newly developed measurement setup that utilizes multiprobe array technique to achieve results in full scanning angle of 3600. A research article explaining the methodology of measurement setup and its working is cited so that readers can refer to the article to have in-depth details about the setup.
Comment no. 7: As suggested, adding current distribution figures for the three layout configurations could provide valuable insights into the differences between them. This visual representation would enrich the understanding of the antenna's behavior.
Author response: The authors would like to thank the respected reviewer for the valuable comment. The current distribution figures of all configurations are added along with a brief explanation. The section is highlighted for reviewer convenience.
Comment no. 8: The novelty of the proposed antenna is limited.
Author response: The proposed paper provides detailed study of two port MIMO antenna with different orientation of antenna element. The antenna has compact size, simple geometry, and low profile. The antenna also offers wideband, high gain and radiation efficiency. The MIMO parameters of proposed antennas are under acceptable range. Moreover, Table 1 also provides the comparison of this work with literature work, which shows that proposed antenna is simple, small, operating over wide bandwidth, having high gain and isolation as compared to other work presented in literature.

Reviewer 2 Report
1. The author must correctly define the antenna size in the introduction section. (12 mm 24 mm by 0.79 mm). They must also check the paper for similar mistakes, like in the conclusion section (high of 11.75 dBi).
2. The author must define the MIMO parameter before using the abbreviation in the introduction section. (CLL)
3. The author must check in sections 3 and 3.1 and write correctly about the S11 definition and MEG. (S11 -10dB.), (Mean Effective gai (MEG))
4. The author must put legend detail along with the curve in Fig.9, Fig.12, and Fig.15.They may also calculate the MEG ratio between antenna elements, which should come to < 3dB.
5. The Author must briefly define MIMO parameters and which formula they used for the calculation. They may provide references for it.
6. The author must cross-verify minimum isolation results from the plots shown in Table 2. They have mentioned maximum isolation instead of minimum value. The plot of the S parameter may also rearrange the axis value for better visibility.
7. It needs to be clarified in the conclusion section which MIMO case they have recommended and provided parameter details.
8. The author may also acknowledge the facility they have utilized for antenna simulation and measurement. The author may give details of the measuring equipment.
the paper needs minor editing.
Author Response
Comment no. 1: The author must correctly define the antenna size in the introduction section. (12 mm 24 mm by 0.79 mm). They must also check the paper for similar mistakes, like in the conclusion section (high of 11.75 dBi).
Author response: Authors are thankful to reviewer for in-depth revision. The author has revised the paper and made possible changes to remove the typos and write the correct size of antenna presented in introduction section. Moreover, the typo mistake of gain presented in conclusion is also updated in revised paper.
Comment no. 2: The author must define the MIMO parameter before using the abbreviation in the introduction section. (CLL)
Author response: Authors appreciate the respected reviewer for constructive comments. In the revised paper, the abbreviation of MIMO parameters is defined properly.
Comment no. 3: The author must check in sections 3 and 3.1 and write correctly about the S11 definition and MEG. (S11 -10dB.), (Mean Effective gai (MEG))
Author response: Authors appreciate the respected reviewer for their time and constructive comments. The author has removed the highlighted typos and updated the revised manuscript.
Comment no. 4: The author must put legend detail along with the curve in Fig.9, Fig.12, and Fig.15.They may also calculate the MEG ratio between antenna elements, which should come to < 3dB.
Author response: The legends of Fig. 9, 12 and 15 have been updated moreover usage of additional indicator is also inserted that will help the reader to easily understand the graphs.
Comment no. 5: The Author must briefly define MIMO parameters and which formula they used for the calculation. They may provide references for it.
Author response: The MIMO parameters are well defined in literature and their formula is also added in a lot of papers, thus, to avoid any sort of extra length of paper we have carefully cited the proper reference where all the MIMO parameters are defined along with their respective formulas.
Comment no. 6: The author must cross-verify minimum isolation results from the plots shown in Table 2. They have mentioned maximum isolation instead of minimum value. The plot of the S parameter may also rearrange the axis value for better visibility.
Author response: Table 2 is updated in the revised manuscript following the suggestion of respected reviewer.
Comment no. 7: It needs to be clarified in the conclusion section which MIMO case they have recommended and provided parameter details.
Author response: Following the constructive comment of the respected reviewer the conclusion has been updated and the updated part is highlighted in the revised manuscript.
Comment no. 8: The author may also acknowledge the facility they have utilized for antenna simulation and measurement. The author may give details of the measuring equipment.
Author response: The author has added in revised paper about the software tool used to design antenna. Moreover, the information about measurement setup is also added in the start of section 3.
Round 2
Reviewer 1 Report
The authors have revised the manuscript according to part of my reviewer comments. Please reply to all my previous and the following reviewer comments one by one.
1. Please add all necessary parameters to show the detail dimension and the coordinate system in Fig.1 and Fig.5.
2. Please add the color map of the current distribution figures in Fig. 16
3. Please add the photo of measurement setup such as the anechoic chamber.
4. What do three lines stand for in Fig. 3?
5. It seems that the measured lowest value is below -50 dB in Fig.8. Please show this lowest value in this Figure.
6. Why the Simulated E-plane and the measured H-plane patterns only cover 0 to 180 degrees in Fig. 5? Please add the co-pol and the cross-polar radiation patterns.
7. Please make sure the radiation efficiency is as high as 96% in Fig.6. Please add the radiation efficiency parameter in Table 1.
8. Please only compare the performance of 2 ports MIMO antennas with this work in Table 1.
9. The authors should include all equations to calculate ECC, MEG, DG, and CCL.
The manuscript contains grammatical errors that necessitate correction.
Reviewer 2 Report
The paper can be accepted.
It is Ok
Author Response
Thank you, and appreciate your feedback
Round 3
Reviewer 1 Report
The authors have revised the manuscipt according to the reviwer comments. The manuscript contains grammatical errors that necessitate correction.
The authors have revised the manuscipt according to the reviwer comments. The manuscript contains grammatical errors that necessitate correction.